# Laser Processed Hybrid Lead-Free Thin Films for SAW Sensors

**DOI:** 10.3390/ma15238452

**Published:** 2022-11-27

**Authors:** Nicoleta Enea, Valentin Ion, Cristian Viespe, Izabela Constantinoiu, Octavian Buiu, Cosmin Romanitan, Nicu Doinel Scarisoreanu

**Affiliations:** 1National Institute for Laser, Plasma and Radiation Physics, 077125 Magurele, Romania; 2Department of Physics and Astronomy, University of Florence, Via G. Sansone 1, 50019 Sesto Fiorentino, FI, Italy; 3Faculty of Physics, University of Bucharest, 077125 Magurele, Romania; 4National Institute for Research and Development in Microtechnologies, 077190 Voluntari, Romania

**Keywords:** barium strontium titanate, polyethylenimine, PLD, MAPLE, SAW device, gas sensor

## Abstract

In this study we report the specific interaction of various gases on the modified surface of acoustic wave devices for gas sensor applications, using the piezoelectric ceramic material BaSrTiO_3_ (BST), with different concentrations of Sr. For enhancing the sensitivity of the sensor, the conductive polymer polyethylenimine (PEI) was deposited on top of BST thin films. Thin films of BST were deposited by pulsed laser deposition (PLD) technique and integrated into a test heterostructure with PEI thin films deposited by matrix assisted pulsed laser evaporation (MAPLE) and interdigital Au electrodes (IDT). Further on, the layered heterostructures were incorporated into surface acoustic wave (SAW) devices, in order to measure the frequency response to various gases (N_2_, CO_2_ and O_2_). The frequency responses of the sensors based on thin films of the piezoelectric material deposited at different pressures were compared with layered structures of PEI/BST, in order to observe differences in the frequency shifts between sensors. The SAW tests performed at room temperature revealed different results based on deposition condition (pressure of oxygen and the percent of strontium in BatiO_3_ structure). Frequency shift responses were obtained for all the tested sensors in the case of a concentration of Sr x = 0.75, for all the analysed gases. The best frequency shifts among all sensors studied was obtained in the case of BST50 polymer sensor for CO_2_ detection.

## 1. Introduction

Nowadays, poor air quality is an important topic to consider because of its hazardous consequences on health, of different air pollutants such as toxic gases (CO, H_2_S, NO_x_, SO_x_), volatile organic compounds (VOCs) and greenhouse gases (CO_2_, CH_4_) which are responsible for serious health issues [1,2].

One of the key components of the air monitoring system is represented by gas sensors due to their extensive applications, both for indoor and outdoor air quality measurement systems, environmental science, medical field, automotive industry and defence. Usually, gas sensors are composed of a transducer and an active layer, which convert a chemical reaction into measurable quantities such as change in resistance, current, voltage or frequency [3]. In terms of detection, there are several types of gas sensors, each with advantages and disadvantages, namely catalytic gas sensors [4]: simple, low cost technology, but requiring air or oxygen to work and can be affected by lead or chlorine; thermal gas sensors [5]: robust and simple, but the reaction is due to heating wire; electrochemical sensor [6]: measures low concentration of toxic gases, but failure modes are unrevealed; IR sensors [7]: use only physical response, but not all gases present absorption in IR domain; and sensors based on acoustic wave propagation [8]. Gas sensors, based on acoustic transduction, can be manufactured in submillimetre size, have high sensitivity and can be produced at low costs. Moreover, acoustic sensors can effortlessly be integrated with wireless communication systems and these features make them into being used in various domains [9,10].

In acoustic devices, a piezoelectric material is used for generating acoustic waves based on piezoelectricity. High frequency vibrations propagate through these devices, either in bulk or on the surface of the material, which produces a perturbation and sequentially an elastic deformation, that translates into an acoustic wave traveling the material. When talking about gas sensing, there are two types of devices that are used commonly, more precise bulk acoustic wave devices (BAW) and surface acoustic wave devices (SAW). In the case of BAW, the wave travels through the bulk of the material, for example [11] quartz crystal microbalance (QCM) and film bulk acoustic resonator (FBAR). In the second type, the acoustic waves are confined to the surface of piezoelectric materials and are known as the surface acoustic wave devices (SAW) [12,13]. There are numerous papers in the literature, including review papers, that compare these two types of sensors (BAW and SAW) and in which their differences and limitations are discussed [9,14]. Basically, SAW sensors use surface acoustic waves, and BAW use bulk or volume acoustic waves. In BAW devices, acoustic waves propagate through the bulk structure of the device active layer. The BAW structure consists of the piezoelectric thin film sandwiched between two metal electrodes and the acoustic waves propagate through the bulk structure of the device active layer. Thickness shear mode (TSM) resonators and one of the common utilized BAW devices. Its widespread use comes from the fact that they have temperature stability, they are easy to fabricate and they are also sensitive to additional mass deposited on the surface [15]. Moreover, these devices can be used as biosensors due to their capability of measuring and detecting liquids. The drawbacks of TSM come from their low mass sensitivity, considering that the range in which they operate is 5–30 MHz. An alternative for increasing their sensitivity would be to produce thinner devices, but that would result into sensors that are fragile and difficult to handle [16]. A sensor has increased sensitivity the more energy in the propagation path that is being perturbed. In the case of BAW sensors, the energy is dispersed from the surface of the sensor, travelling the bulk and to the other side, leading to a low density of energy on surface. The surface acoustic wave sensors (SAW) works by having the energy concentrated on the surface, thus increasing its sensitivity [9]. The SAW sensor typically consists of a piezoelectric thin film with two metal electrodes on the same side of the piezoelectric substrate. The acoustic wave propagates on the surface of the piezoelectric surface between these two electrodes (input and output).

One of the extensively studied piezoelectric materials used in gas sensing SAW devices is barium titanate [17,18,19], as an alternative to replace toxic lead-based titanium compounds (PbZrTiO_3_-PZT), considering that it exhibits dielectric, photoelectric, ferroelectric and piezoelectric properties, being the first discovered ferroelectric perovskite oxide material [20,21,22]. Depending on the temperature, BaTiO_3_ has four polymorph phases (cubic, tetragonal, orthorhombic and rhombohedral, consecutively). However, at room temperature, it is tetragonal in structure [23]. The ferroelectric phase is formed when the temperature decreases, through several phase transitions: cubic (centrosymmetric phase) to tetragonal, orthorhombic and rhombohedral phases. All the previously mentioned phases (excepting the cubic phase) show ferroelectric properties. The displacement of Ti^4+^ cations and the orientation of the formed spontaneous polarizations are along [001], [101] and [111] directions [23,24]. Moreover, besides electric properties, BaTiO_3_ has increased thermal and chemical stability and can be used in large application domains. The Curie temperature of BaTiO_3_ is 120 °C, above this temperature the material is in paraelectric state and has a cubic structure stable up to 1460 °C [25]. Solid solutions resulting from the barium replacement with other elements, lead to an improvement in the applicative potential of these materials in various fields, by modifying the dielectric properties. For applications in the radio frequency (RF) and microwave field, but also in the optical range, one of the requirements is performing in paraelectric state, because its electrical permittivity does not present thermal hysteresis and has small losses [26]. One of the substituents of the barium is Sr^2+^, so the substitution of the Ba^2+^ ions with the Sr^2+^ ions leads to a monotone decrease in the transition temperature between the ferroelectric and para-electric phases [27]. Studies of variation in the percentage of Sr for Ba_1−x_ Sr_x_TiO_3_ compositions showed that for x = 0.95, the Curie point is 40 K. The tetragonal-orthorhombic transition is not modified, but the orthorhombic-rhombohedral temperature transition will decrease linearly with the increase in the Sr molar fraction. Barium strontium and titanate, Ba_1−x_Sr_x_TiO_3_ or abbreviated BST, is the most intensely studied material over the past 20 years, with a wide range of use on almost the entire frequency spectrum in low-frequency electronic circuits and microwaves [28,29,30,31]. 

BST is an important material for tuneable microwave devices due to its high dielectric constant, large electric field turnabilities, low dielectric loss and variable Curie temperature (from 30 to 400 K), depending on the strontium doping [32]. It has been used for microwave device applications such as phase shifters, tuneable filters, delay lines and tuneable oscillators [33]. At the same time, BST thin films have emerged as a viable platform for multimodal sensing functionalities for instance pressure, force, temperature, humidity, gas detection and IR radiation, due to their high capacitance, switchable spontaneous polarization, pyroelectric, piezoelectric, photovoltaic and electro-optic effects [34,35,36,37,38].

In this study, we present the specific interaction of various gases on the modified surface (physically or chemically) of surface acoustic wave devices for gas sensor applications using BaSrTiO_3_ (BST) thin films. For enhancing the sensitivity of the sensor, a conductive polymer polyethylenimine (PEI) thin film was deposited on top of the BST thin films. Conducting polymers are organic materials that present a high resistivity towards external stimuli. Among these materials, PEI [39,40] has attracted wide interest because of the versatility in its use, easy synthesis, good environmental stability, together with a favourable response at room temperature to guest molecules [41]. Thin films of BST were deposited by pulsed laser deposition technique, interdigital electrodes (IDT) were further deposited on top of the thin films by thermal evaporation, followed by PEI thin film deposition using matrix assisted pulsed laser evaporation. Further on, the layered structures were integrated in surface acoustic wave devices, in order to measure the frequency response to various gases (N_2_, CO_2_ and O_2_). The frequency responses of the sensors based on thin films of the piezoelectric material BST deposited at different pressures were compared with layered structures of PEI/BST in order to observe differences in the frequency shifts between sensors.

After our knowledge, there are no reports of using laser deposition techniques for depositing layered structures of PEI/BST and integrating them into surface acoustic wave devices for gas sensors. 

## 2. Materials and Methods

### 2.1. Thin Films and IDT Deposition

The Ba_1−x_Sr_x_TiO_3_ (BST) thin films were deposited by pulsed laser deposition (PLD) using two different composition of Sr concentrations, as follows: x = 0.5 and x = 0.75, with the purpose of evaluating how the composition influences the gas sensing response. The BST_x_ targets were prepared by conventional sintering route [42]. The substrates used for deposition were of alpha-alumina (Al_2_O_3_), with the crystal structure of corundum and the lattice constants a = 4.785 and c = 12.991 [43]. For the fabrication of the BST/Al_2_O_3_ samples a Nd:YAG (266 nm) laser was used, the laser fluence value being 1.65 J/cm^2^. For all samples, the number of laser pulses was set at 36,000, with a 10 Hz repetition rate. The substrate temperature was set at 700 °C and the heating rate between room temperature (RT) and deposition temperature was 50 °C/min, while the cooling rate was 10 °C/min. The flow of oxygen during ablation process was set at 35 sccm and the gas pressure reached 1 × 10^−1^ mbar or 2 × 10^−1^ mbar, before starting the PLD process. When the ablation process ended, the cooling of substrates was performed in oxygen flow, set at 200 sccm. For each composition of BST and oxygen pressure, we deposited four probes in a row, and we checked their morphological and structural properties before we proceeded with the deposition of gold electrodes, followed by gas response measurements. 

The conductive polymer polyethylenimine (PEI) was deposited using matrix assisted pulsed laser evaporation (MAPLE). This deposition method was chosen taking into consideration that polymers are sensible to high temperatures and this deposition technique allows the deposition of various organic and inorganic materials without destroying the structure and maintaining the stoichiometry and intrinsic characteristics of the material. The MAPLE deposition method involves obtaining a target formed from the material to be deposited (1–5 wt. %) dissolved in a solvent (matrix). The solvent is chosen so that the material to be deposited can dissolve (without chemical interactions) and the laser energy is absorbed by the solvent, and not by the material to be deposited. Thus, a layer is formed, that originates from the evaporated polymer molecules, while the volatile solvent molecules are evacuated through the pump from the deposition chamber. The PEI target was obtained by dissolving 400 mg of PEI in a 1:1 solution of deionized water and isopropyl alcohol, under magnetic stirring for 60 min. The target thus obtained was frozen in liquid nitrogen and further on irradiated, using the fourth harmonic (266 nm) of a Nd:YAG laser, at 10 Hz with 36,000 laser pulses. The thickness of the deposited films was of 298 nm with roughness of 120 nm.

To evaluate the gas response of the studied materials, interdigital electrodes were deposited on the surface of thin films, by thermal evaporation with the purpose of integrating them into surface acoustic wave sensors. In a SAW sensor, the electrical signal applied on the input interdigital transducers (IDTs) is converted into mechanical energy that generates surface acoustic waves. These waves, characterized by a certain frequency, cross the surface of the sensor to the other pair of IDTs (output). The output IDTs convert the mechanical energy in an electrical signal, with characteristics depending on the proprieties of the arriving wave front. If on the way to the second pair of IDTs, the wavefront encounters different disturbing factors, it changes its oscillation frequency. Among the disturbing factors are the gas molecules that can appear at the level of the sensitive film and that, through mass accumulation or due to some acoustic–electrical interactions, change the oscillation frequency of the wave front, a response that this current research is being emphasized on [44,45]. For the IDT, Au metal electrodes were deposited, with thicknesses of approximately 200 nm on the piezoelectrically active thin layers, using an interdigital electrode mask with a period of 50 μm and a digit width of 50 μm. For the good adhesion of the gold, an initial layer of chrome was deposited, with a thickness of 10–20 nm, both thermal evaporations being carried out successively. The thicknesses of the two deposited layers were monitored in situ with the help of a calibrated quartz crystal. Later, gold wires with a diameter of 75 µm were glued to the electrodes, using conductive silver (Ag) paste, with Ag concentration of at least 70%. The sensors were tested and characterized for different types of gases (O_2_, CO_2_ and N_2_), at room temperature. 

Using the methods and techniques described above, the first evaluation of gas response of BSTx was conducted, followed by a second deposition in which we covered the BSTx samples with the sensitive polymer (PEI) using the MAPLE technique. Afterwards, we proceeded with the integration of the samples in the SAW device and subsequently, we tested them a second time, in order to investigate any changes in the sensor response.

### 2.2. Sample Characterization and SAW Measurements

Surface morphological characterization of the deposited thin films was performed using scanning electron microscopy (SEM) (JSM-531 Inspect S Electron Scanning Microscope, FEI Company, Prague, Czech Republic) and also atomic force microscopy (AFM) technique (model XE100, Park Systems, Suwon, Republic of Korea) working in non-contact mode and recording the topography of 20 × 20 μm^2^, thus allowing a good sample viewing and positioning.

The investigation of optical properties was performed using reflection type spectroscopic ellipsometry measurements, on a Woollam Variable Angle Spectroscopic Ellipsometer (VASE) system equipped with a high-pressure Xe discharge lamp, in the spectral range of 1–5 eV, from the near IR to the UV (300–1200 nm) and were carried out at a fixed angle of incidence. The change in the polarization state of linearly polarized light due to reflection at the surface was measured. In order to obtain refractive indexes and extinction coefficients for the thin films, WVASE32 software (VASE, J.A. Woollam Co., Inc., Lincoln, NE, USA) was used for fitting and extracting useful data from complex multilayer response. 

High-resolution X-ray diffraction analysis was performed using a 9 kW Rigaku SmartLab diffractometer (Osaka, Japan) operated at 45 kV and 200 mA. During the measurements, the incidence angle was fixed at 0.5°, while the detector scanned from 2θ = 20–70°. The incidence and receive slits were set accordingly to reduce the instrumental broadening associated with the optical components. 

The SAW frequency response testing system was composed of an amplifier, a frequency counter, a mass flow meter, a mass flow controller and a computer. The signal transmitted by the sensor was amplified by DHPVA-200 FEMTO amplifier (Messtechnik GmbH, Berlin, Germany) and the frequency jump was recorded by a CNT-91 Pendulum frequency counter (Spectracom Corp, Rochester, NY, USA) connected to a computer software, Time View 3. The gas flow was controlled using a system consisting of a gas flow controller and a flow meter connected to the test gas cylinder. The gas concentration employed for testing was: N_2_–99.996%, O_2_–99.999% and CO_2_–99.998%. For all determinations, the total gas flow rate was maintained constant at 0.5 l/min. All measurements were performed at room temperature. 

## 3. Results and Discussion

### 3.1. AFM and SEM Characterization

The morphology of the BST samples grown on Al_2_O_3_, obtained in the same conditions (the O_2_ working pressure of 0.1 and 0.2 mbar, temperature of 700 °C, distance from the target of 4 cm and laser fluence of 1.65 J/cm^2^ for PLD and 0.5 J/cm^2^ laser fluence for MAPLE depositions) shows that the deposited thin films did not present major defects or cracks, with the presence of grains on surface of BSTx thin film. The grains were found for both oxygen deposition pressure and the concentration of strontium. The surface of all analysed samples was fully covered by BSTx and the roughness were found to be tens of nanometres (50–80 nm) (Figure 1). 

The images obtained by the SEM technique (Figure 2) were in accordance with AFM results, showing a uniformly distribution of the materials on the surface, for all depositions. In the case of BST50 ((a) and (b)), for both working pressures of 0.1 and 0.2 mbar, the film was composed of well-defined randomly orientated crystallites, scattered uniformly, compared to BST75, where the crystallites are gathered into clusters, but still uniformly distributed. In the case of the latter, some small defects can be observed in the surface formation of the crystallites.

### 3.2. XRD Measurements 

Grazing-incidence XRD investigations reveal the presence of a couple of diffraction peaks for each value of Sr content and O_2_ pressure, as shown in Figure 3.

The diffraction peak indexing was achieved using the ICDD (International Centre for Diffraction Data) database, which shows the presence of alumina substrate (card no. 002-1373), as well as of the perovskite barium strontium titanate (BST) according to card no. 034-0411. A careful investigation in the angular range of 30–34° indicates that the main diffraction peak of BST compound associated with (110) reflection was shifted along different Sr content and O_2_ pressure. The angular units (2θ) were converted in the interplanar distance, $d_{hkl}$ values with Bragg’s law, thus: 2dsinθ=λ, where λ is the incident wavelength (1.5406 Å). Then, the standard relation for cubic crystal was applied to obtain the lattice constant a [46]. Moreover, the diffraction peaks were fitted with a pseudo-Voigt fit, that provided the position, as well as the peak broadening. Accordingly, it was obtained that the lattice constant, a varied as: 3.95 Å (BST50_0.2 mbar), 3.91 Å (BST75_0.2 mbar), 3.97 Å (BST50_0.1 mbar) and 3.91 Å (BST75_0.1 mbar). In order to calculate the effective concentration of Sr, x in our quaternary compounds, it is necessary to apply the Vegard’s law, that assumes a linear relation between unit cell length a and stoichiometry in Ba_1−x_Sr_x_TiO_3_ [47].
(1)$$a_{Ba_{1-x}Sr_{x}TiO_{3}}\left(x\right)=a_{SrTiO_{3}}x+a_{BaTiO_{3}}\left(1-x\right)$$
where $a_{SrTiO_{3}}$ = 3.095 Å and $a_{BaTiO_{3}}$ = 4.001 Å.

Based on the Vegard’s law and the experimental value of $a_{Ba_{1-x}Sr_{x}TiO_{3}}$ from GIXRD, one can obtain the relative Sr content was: 0.86 (BST75 at both O_2_ pressure), 0.42 (BST50_0.2mbar) and 0.26 (BST50_0.1mbar). At the same time, it seems that the peak broadening was affected at different Sr contents/O_2_ pressure, as it is reflected from the values of the FWHM (Full Width at Half Maximum). This was evaluated from the pseudo-Voigt fit as the sum between the Gaussian and Lorentzian component, and following values were obtained: 0.391° (BST50_0.2 mbar), 0.357˚ (BST75_0.2 mbar), 0.414˚ (BST50_0.1 mbar) and 0.435˚ (BST75_0.1 mbar). Different values of FWHM, β further imply different sizes of the crystalline domains (also called mean crystallite size, τ), whose values were assessed using the Scherrer equation [48]:(2)τ=kλβcosθ
where k is the shape factor of the crystalline domains, usually taken as 0.93. Accordingly, the following values were obtained: 21.8 nm, 23.9 nm, 20.6 nm and 19.6 nm, being revealed in fact that the crystal quality becomes better (i.e., dislocation density is smaller) at higher O_2_ pressure. Moreover, comparable values for the mean crystallite size were obtained for barium titanate obtained by sol-gel method [49,50]. Overall, the structural investigations based on XRD confirm the presence of the different Sr concentrations in BaTiO_3_ lattice and suggest that the higher O_2_ pressure could enhance the crystal quality. 

### 3.3. Optical Properties: Spectrometric Ellipsometry (SE)

The thicknesses and the optical constants dispersion of BSTx samples were calculated from spectro-ellipsometry. The measured spectra of ψ and Δ, which are usually measured parameters in ellipsometry were fitted using an optical model composed of a stack of three layers: the Al_2_O_3_ substrate, the BSTx layer and the top rough layer. The dielectric function of Al_2_O_3_ layer (or to optical constants dispersion) was taken from Woollam ellipsometer database. For the BSTx layer the dielectric function was determined by fitting the experimental data by a single Gauss oscillator. For the top rough layer, we used a mixture of BSTx and voids. The mixture was approximated to have a composition of 50% BST and 50% voids in Bruggeman approximation. The fit quality was given by a small value of MSE (mean square error) [51]. The BSTx layer thickness values and the optical dispersion are presented below in Table 1 and Figure 4.

In the case of BST50 samples: the thickness of samples was found to be ~390 nm for the lower pressure of oxygen (0.1 mbar) and 190 nm for higher pressure. The difference between these two values can be explained by the effect of the confined plasma plume during the ablation process at higher pressure. The roughness calculated on an area of ~10 mm^2^ was between 25 and 60 nm, similar to the roughness evidenced by AFM images.

The refractive index dispersion for the BST50 (Figure 4) indicates a strong dependence on deposition parameters (in our case the oxygen pressure). The value of “n” at λ = 600 nm was n~2.2 for the sample growth at 0.1 mbar of O_2_ and this value decreased to n~2.07 for the 0.2 mbar of O_2_. The dependence of refractive index with the oxygen pressure was reported by Liu et al. [52], and they evidenced the strong dependence of “n” with O_2_ pressure. In term of optical absorption (or values of extinction coefficients “k”), for all samples of BST50 the “k” starts to increase below 400 nm (or at photon energy higher than 3.1 eV) and those values are quite normal for barium titanate doped with strontium. According with Boubaia et al. [53], the BSTx exhibit first critical points at energies 3.08–3.2 eV for BST50 and BST75 due to direct optical transitions between the VBhigh and CBlowest [53]. At photon energies lower than 3 eV the BST is optical transparent. 

For the BST75 sample, the thickness for the thin film deposited at a pressure of 0.1 mbar O_2_ was found to be ~300 nm, lower than BST50, even if the deposition parameters were the same. The difference can be explained by different laser absorption characteristics of the ceramic targets, due the presence of different amount of strontium.

The values of “n” for the sample obtained in lower pressure are slightly higher than samples obtained in 0.2 mbar of O_2_ (n~2.2 for 0.1 mbar and n~2.15 for 0.2 mbar). The behaviour of extinction coefficients for BST75 is quite similar with BST50 for a pressure of 0.1 mbar O_2_ but, when the pressure is increased, the extinction coefficients start to increase at lower photon energy (λ~450 nm or 2.75 eV). According with Boubaia [53], the photon energy where the optical absorption occurs should be at 3 eV. In our case, the lower values of photon energy, where the “k” starts to increase indicate the presence of defects.

From spectro-ellipsometry analysis, all samples of BSTx deposited by laser ablation technique exhibit a high refractive index (n > 2) and optical transparency at photon energies lower than 3 eV. The calculated roughness is similar with the AFM results. 

### 3.4. SAW Measurements for Different Gases

As previously mentioned, the performance of both BST and the layered heterostructure of PEI/BST was evaluated for the detection of several potential toxic gases. In order to achieve this, firstly Au/Cr electrodes were deposited on all BST/Al_2_O_3_ thin films as presented in the Materials and Methods section. Further on, samples of BST50/75 obtained at both working pressures of 0.1 and 0.2 mbar O_2_ were used in the MAPLE deposition of PEI, with the mention that the side contacts of the electrodes were protected during the polymer deposition to prevent them from being covered by the PEI thin film. 

The tested sensors with their characteristics are presented in Table 2. The measurements were performed on sensors with and without sensitive polymer layer polyethyleneimine (PEI). The tests were carried out for nitrogen (N_2_), oxygen (O_2_) and carbon dioxide (CO_2_). The testing system is presented in Figure 5. The frequency shift for all tested probes can be found in Table 2. For the sensors that yield a frequency response, we performed around 10 measurements and we found the repeatability of the response had differences of below 3%, with a maximum difference in reproducibility of 5%.

Figure 6 shows results for the BST50_0.1 mbar O_2_ and BST50_0.2 mbar O_2_ sensors, both with polymer layer and without polymer layer. 

The mechanism of gas sensing was explained by Wang et al. for the oxide materials [54]. When gas molecules are interacting with surface of BaSrTiO_3_, the charge state is altered and the conductivity of the sample is changed. The changing in conductivity of layer leads to change in SAW sensor response [55]. The selectivity of SAW sensors dependents entirely upon the sensing layer material properties [56,57]. When a SAW sensor is exposed to a mixture of gases, it absorbs and reacts to them differently, leading to a difference in the intensity of the response for each component. The interaction between the sensing layer and the gas components, thus the response time of a SAW sensor is influenced by several factors. When looking into mass-based sensing films, the response and recovery time strongly depends on the rate of diffusion of the absorbed molecules into the layer and the piezoelectric substrate and back to the film surface. Thinner sensing layers usually have faster response times. By decreasing the thickness, the time required to reach to the equilibrium decreases, considering that the gas diffuses in and out of the film rapidly. Thus, using thin sensing films will result in rapid responses to gases [58,59,60].

The BST50 sensor deposited in O_2_ atmosphere with a pressure of 0.1 mbar did not respond to any of the three gases tested, even after polymer was deposited on the surface of the sensor (Figure 6a). In the case of BST50 sensor obtained in 0.2 mbar oxygen without polymer, there was no response to the tests (Figure 6b), while for those with PEI a response was registered to all gases tested (Figure 6c–e). From Table 2, it can be seen that the best frequency shift among all sensors studied was obtained by the BST 50_0.2 mbar polymer sensor, 0.3 MHz for CO_2_ detection. 

The BST75 deposited at 0.2 mbar O_2_ is the only one that had results for all the tests carried out, both with polymer and without polymer. However, Table 2 shows considerable differences in the frequency shifts between sensors with polymer and those without polymer, for all gases tested. Moreover, in the case of tests without polymer, in the graphs of frequency shifts for BST75 (0.1 mbar O_2_) presented in Figure 7a–c, a noise level of approximately 0.3 kHz was recorded. In the case of sensors with polymer, this noise level was attenuated, and the shifts are better defined. The best frequency shift in the case of this sensor was 0.183 MHz, obtained with PEI polymer, for O_2_ detection.

Comparing the results in the Table 2 and Figure 7 and Figure 8, it was noticed that the BST75 deposited at pressure of 0.1 mbar O_2_ sensor has results only when it was used with PEI polymer (Figure 7b,d,f). Another aspect observed is related to the fact that the frequency shift of the PEI/BST75 at 0.2 mbar sensor, compared to those of the PEI/BST75 (0.1 mbar) sensor, for the same gases, was higher, even tens of times higher for the O_2_ test (Table 2). In the case of the PEI/BST75 at 0.1 mbar sensor with polymer, the best result was registered for the test performed with N_2_, with a frequency shift of 0.068 MHz, as can be observed in Figure 8.

The SAW sensor consists of a sensing film and a conversion element, but the sensing membrane or the sensing film constitutes the most important element of the SAW sensor. The sensing layer acts as a bridge, the target gas molecules being absorbed on the surface of the sensing membrane. The mass loading of the analysed gas molecules (O_2_, N_2_ or CO_2_) corresponds to the shifts in frequency that leads to the detection process. Polymers or carbon-based sensing film are usually used as membranes. In our case, we used the PEI polymer. Choosing this polymer was based on two important aspects: first of all, PEI is easily processable by laser techniques [61]; and secondly, it preserves its properties after laser processing, making it a well-known polymer for gas detection [62]. 

For the single thin film of BST50, deposited at 0.1 and 0.2 mbar O_2_, there was not a clear response in frequency, when gas was inserted in the experimental chamber. The single signal of those sensors was obtained in the presence of nitrogen (Figure 6b), but the signal was accompanied by noise, with an unclear shape. With the deposition of PEI on the surface of BST50, a good signal was registered only for BST50 deposited at 0.2 mbar of O_2_. From XRD analysis, we evidenced the effect of O_2_ pressure during deposition and the results indicated that the crystal quality becomes better at higher deposition pressures, as the dislocations/defects density is lower. This can be an explanation for the overall response in case of BST50 deposited at 0.2 mbar accompanied by the PEI layer. In the case of BST50 obtained at 0.1 mbar, the crystalline disorder suppressed the gas response, even when covered with sensitive PEI film. Moreover, based on the Vegard’s law and the experimental value of lattice “a” calculated from XRD, the relative Sr contents in our samples varied for 0.1 and 0.2 mbar deposition pressures, with Sr 0.42 in the case of BST50_0.2mbar and 0.26 for BST50_0.1mbar. The piezoelectric properties of BST are strongly dependent on amount of Strontium in BaTiO_3_ structure. A value of d33 = 50 pC/N was reported by Ahmed et al. [63] for the BST40 (x = 0.4) and d33 = 377 pC/N for BST80 (x = 0.8). 

In the case of BST75, the Sr content for both deposition pressures was Sr = 0.86 and in this case the layers presented a much more pronounced piezoelectric effect compared to samples of BST50. If we take into consideration only the effect of Sr, we should obtain a good response in gas detection for all BST75 samples, but from experimental data (Figure 7), a noise in frequency response was registered; in the case of BST75_0.1 mbar without a sensitive PEI layer. After the deposition pressure was increased, the curve shape in frequency response, in the presence of all analysed gases (O_2_, N_2_, CO_2_), presented a lower noise. A higher amount of oxygen in the reaction chamber leads to, first of all, an improved crystalline quality of BST, with the highest crystalline size of 23.9 nm obtained for BST75_0.2 mbar and secondly, an enhanced response to all gases tested.

The frequency response of BST75_0.2mbar in the presence of all analysed gasses with different frequency shifts indicated the possibility of having a selective gas detection (O_2_, N_2_, CO_2_) without a membrane layer. The frequency shift was higher when the BSTx was covered by a thin layer of sensing polymer membrane.

## 4. Conclusions

This study demonstrates the feasibility of obtaining BST and layered structures of PEI/BST by using laser deposition techniques (PLD and MAPLE). Two different strontium concentrations and two different deposition pressures of O_2_ were used. Further on, the layered structures were integrated into surface acoustic wave devices, in order to measure the frequency response to various gases (N_2_, CO_2_ and O_2_). The frequency responses of the sensors based on thin films of the piezoelectric material BST were compared with layered structures of PEI/BST in order to observe differences in the frequency shifts between sensors. The SAW tests performed at room temperature revealed different results based on deposition conditions (pressure of oxygen and the percent of strontium in BaTiO_3_ structure). The highest frequency shift among all sensors studied was obtained for BST 50_0.2 mbar polymer sensor, 0.3 MHz for CO_2_ detection. Frequency shift responses were obtained for all tested sensors in the case of a higher Sr concentration in the films, both with and without polymer, obtained at 0.2 mbar oxygen pressure, for all the analysed gases.

## Figures and Tables

**Figure 1 materials-15-08452-f001:**
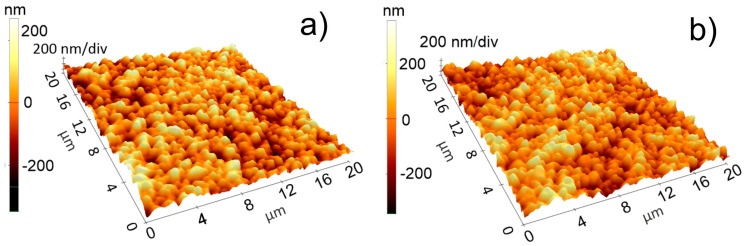
Atomic force microscopy (AFM) images for BST thin films deposited by pulsed laser deposition (PLD) on Al_2_O_3_ heated at 700 °C (**a**) BST50 _0.1 mbar; (**b**) BST75_0.1 mbar.

**Figure 2 materials-15-08452-f002:**
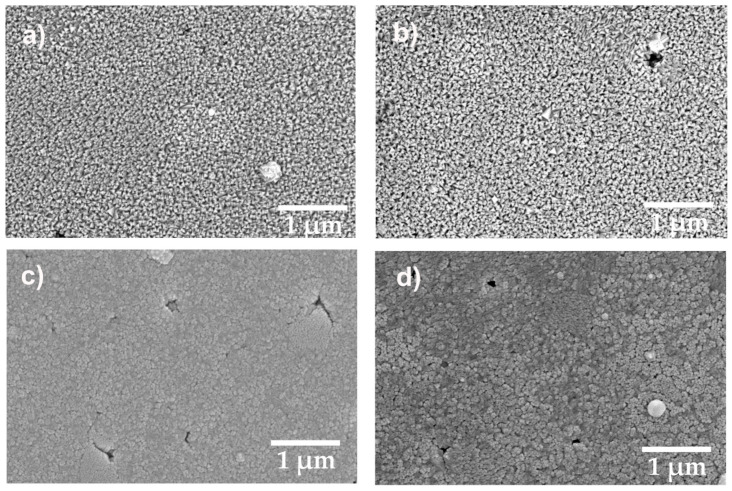
SEM images for BSTx thin films deposited by PLD on Al_2_O_3_ at 0.1 mbar and 0.2 mbar of oxygen pressure; (**a**) BST50/0.1 mbar; (**b**) BST50/0.2 mbar; (**c**) BST75/0.1 mbar; (**d**) BST75/0.2 mbar.

**Figure 3 materials-15-08452-f003:**
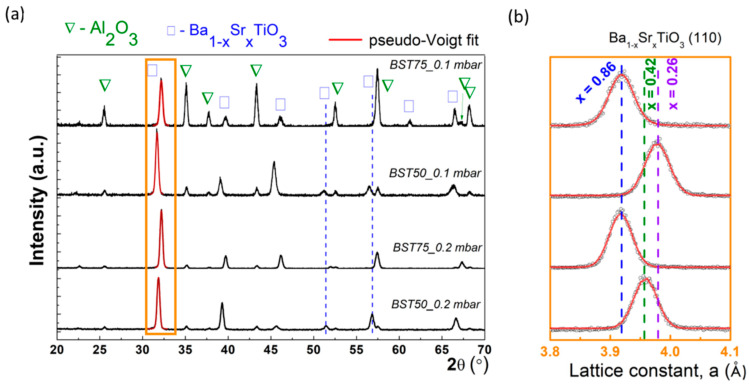
XRD investigation (**a**) Grazing incidence XRD patterns for BST samples at different Sr relative content and O_2_ pressure; (**b**) Evolution of the unit cell parameter, a and the pseudo-Voigt fit for (110) diffraction peak.

**Figure 4 materials-15-08452-f004:**
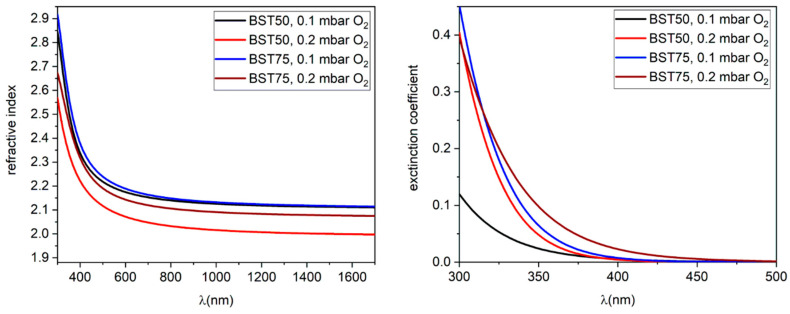
The optical constants measured by spectroelipsometry for BST50 and BST75 samples deposited by PLD in at different oxygen pressures.

**Figure 5 materials-15-08452-f005:**
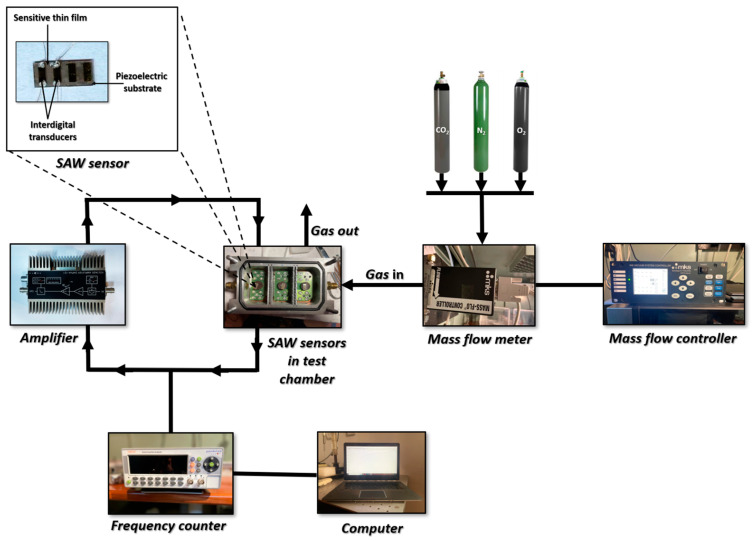
The experimental arrangement used for testing the thin film gas sensors based on SAW device.

**Figure 6 materials-15-08452-f006:**
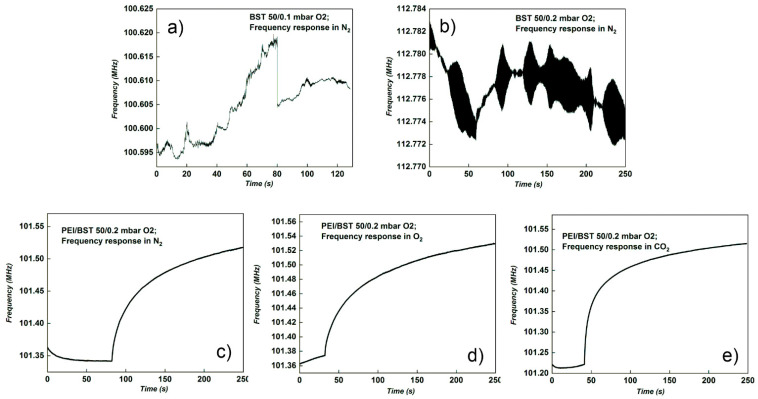
Frequency response of BST50 sensors deposited at 0.1 mbar and 0.2 mbar O_2_ pressure on Al_2_O_3_ gas with and without sensitive polymer (PEI) layer: (**a**) BST50_0.1 to N_2_; (**b**) BST50_0.2 to N_2_; (**c**) PEI/BST50_0.2 to N_2_; (**d**) PEI/BST50_0.2 to O_2_ and (**e**) PEI/BST50_0.2 to CO_2_.

**Figure 7 materials-15-08452-f007:**
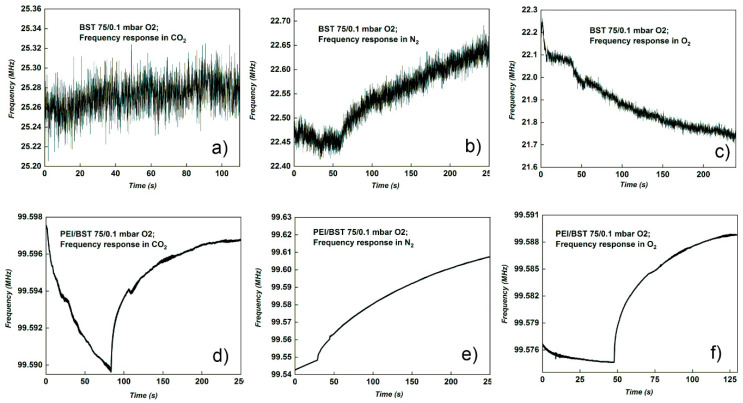
Frequency response of BST75 sensors deposited at 0.1 mbar O_2_ pressure on Al_2_O_3_ gas with and without sensitive polymer (PEI) layer: (**a**)BST75 to CO_2_; (**b**) PEI/BST75 to CO_2_; (**c**) BST75 to N_2_; (**d**) PEI/BST75 to N_2_; (**e**) BST75 to O_2_ and (**f**) PEI/BST75 to O_2_.

**Figure 8 materials-15-08452-f008:**
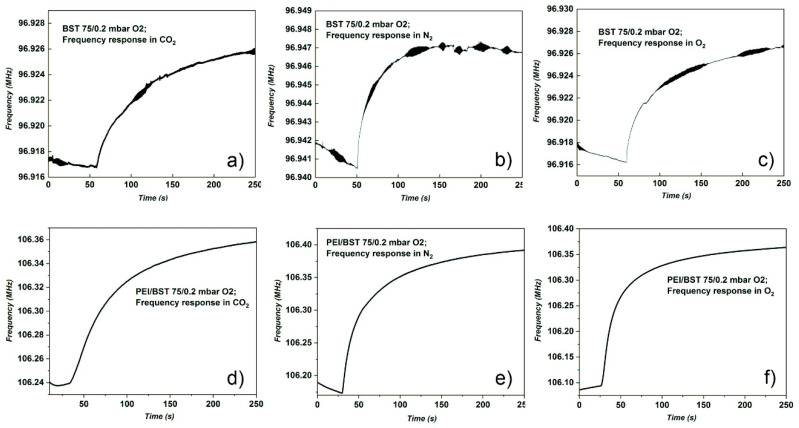
Frequency response of BST75 sensors deposited at 0.2 mbar O_2_ pressure on Al_2_O_3_ gas with and without sensitive polymer (PEI) layer: (**a**) BST75 to CO_2_; (**b**) PEI/BST75 to CO_2_; (**c**) BST75 to N_2_; (**d**) PEI/BST75 to N_2_; (**e**) BST75 to O_2_ and (**f**) PEI/BST75 to O_2_.

**Table 1 materials-15-08452-t001:** Parameters resulting from fitting the SE experimental data for thin perovskite layers deposited by PLD (Amp, En and Br are the parameters of the Gaussian oscillator that describes the dielectric function of the layers, E1 offset is the real part of the dielectric function at infinite frequency).

Probe	O_2_ (mbar)	Thickness (nm)	Roughness (nm)	Amp Gauss	En (eV)	Br (eV)	E1 Offset	MSE
BST50/Al_2_O_3_	0.1	397.1	56.2	7.52	4.97	1.28	3.09	5.487
BST50/Al_2_O_3_	0.2	187.6	34.8	10.12	6.33	2.19	1.52	2.3
BST75/Al_2_O_3_	0.1	302.6	27.2	8.38	5.01	1.36	2.86	5.519
BST75/Al_2_O_3_	0.2	283.7	25.4	6.26	5.23	1.76	2.81	5.112

**Table 2 materials-15-08452-t002:** Sensors with their synthesis characteristics and the frequency shifts of the sensors tested at room temperature for N_2_, O_2_ and CO_2_.

Sensor	Frequency Shift (MHz)					
	N_2_		O_2_		CO_2_	
	Without Polymer	With Polymer	Without Polymer	With Polymer	Without Polymer	With Polymer
BST50 0.1 mbar	-	-	-	-	-	-
BST50 0.2 mbar	-	0.187	-	0.183	-	0.3
BST75 0.2 mbar	0.0066	0.224	0.0017	0.270	0.01	0.12
BST75 0.1 mbar	-	0.068	-	0.0025	-	0.0072

## Data Availability

Not applicable.

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
