# Peer review of "Laser Processed Hybrid Lead-Free Thin Films for SAW Sensors"

_materials, 2022, doi:10.3390/ma15238452_

Round 1

Reviewer 1 Report

In the manuscript titled “Laser processed hybrid lead-free thin films for SAW sensors” (Manuscript ID: materials-2050303), the authors report on the sensing properties of acoustic wave devices based on hybrid polyethylenimine/BaSrTiO3 sensing layers for N2, O2, and CO2 detection at room temperature. The sensing measurements were conducted in pure N2, O2, and CO2 instead of low concentration, which does not provide attractive results in terms of application. Also, the cross-sensitivity effect of humidity was not studied. Several questions should be addressed before publication in Materials Journal. Therefore, I recommend major revision. Please, find below the main questions and comments for the authors:

1.      The manuscript presents some sentences that need to be clarified and also several mistakes concerning English grammar. Please, double-check the manuscript in detail.

2.      AFM, SEM, and Ellipsometry analyses were presented only for the BST samples (without the PEI layer). However, sensing results revealed that the hybrid PEI/BST samples are the most interesting materials for gas detection. Therefore, their surface characteristics should be provided from AFM and SEM studies and correlated to their sensing response.

3.      Table 1 shows that the thickness of films deposited using 0.1 mbar decreases by increasing the Sr amount (x=0.5 to 0.75). However, the opposite behavior occurs for the films deposited using 0.2 mbar. These results should be discussed.

4.      How many measurements were performed in the same device to analyze the repeatability of the response? Also, how many sensors were tested to examine the reproducibility of results? 

5.      What is the proposed sensing mechanism that explains the higher response of BST50 0.2 mbar to CO2 and the selective detection of BST75 0.2 mbar considering the effect of the polymer layer?

6.      There are some differences between the discussion and the Figures and Tables. For example, the authors mentioned noise measurements in Figure 7 (a), (c), and (e). However, this was observed in Figures 7(a), (b), and (c). Also, on page 12, the authors discussed the results of the BST75 sample, and the higher response of 0.183MHz mentioned is related to the BST50 sample, as observed in Table 1. The explanation of the results should be clear.

7.      The highest response obtained was from the BST50 0.2 mbar sample (0.3 MHz frequency shift for CO2), which is different from the statement in the last sentence of the abstract. It should be corrected.

8.      Both Sr contents for the BTS50 samples were attributed to the sample prepared in 0.2mbar. It should be corrected (pag.7). 

9.      References or a brief description of the sintering route used for target preparation should be included in section 2.1.

10.   The X-axis in Figures 6a-b should be labeled.

11.   The abbreviations QCM and FBAR on page 2 should be defined for a non-specialized audience.

Author Response

Response to Reviewer 1

In the manuscript titled “Laser processed hybrid lead-free thin films for SAW sensors” (Manuscript ID: materials-2050303), the authors report on the sensing properties of acoustic wave devices based on hybrid polyethylenimine/BaSrTiO3 sensing layers for N2, O2, and CO2 detection at room temperature. The sensing measurements were conducted in pure N2, O2, and CO2 instead of low concentration, which does not provide attractive results in terms of application. Also, the cross-sensitivity effect of humidity was not studied. Several questions should be addressed before publication in Materials Journal. Therefore, I recommend major revision. Please, find below the main questions and comments for the authors:

Point 1: The manuscript presents some sentences that need to be clarified and also several mistakes concerning English grammar. Please, double-check the manuscript in detail.

Response 1: Thank to reviewer for carefully reading our manuscript. We have double-checked the manuscript and the mistakes were corrected.

Point 2:  AFM, SEM, and Ellipsometry analyses were presented only for the BST samples (without the PEI layer). However, sensing results revealed that the hybrid PEI/BST samples are the most interesting materials for gas detection. Therefore, their surface characteristics should be provided from AFM and SEM studies and correlated to their sensing response.

Response 2: Indeed, for the hybrid PEI/BST samples we obtained better results for gas detection. We used PEI as top layer to enhance the sensitivity of the sensor because PEI is a well-known material for gas sensing, especially for CO2 detection .The novel results presented in our paper, are on simple BSTx sensing layer without polymer. For example, for BST50 deposited at different oxygen pressures (0.1 and 0.2 mbar) did not respond in frequency when gases were introduced in the experimental chamber or their response was very noisy. When the BST50 were covered with PEI, the gas response became, as we expected, very clear and this is a normal behaviour because PEI layer absorbs gas molecules. The samples of BST75 without polymer responded to gas, even if the measured frequency presented some noises.

We did not present a study of PEI because some of co-authors have published these studies in the past and they fully characterized PEI layers obtained by MAPLE. In our work, we used the same previously demonstrated route to obtain PEI thin films. Because this is an experimental approach, we tried to avoid any unexpected experimental results in MAPLE depositions and we deposited all BST samples at once, during one deposition process. This was possible because we used a 2-inch frozen target (PEI + solvent) and a 2-inch sample holder positioned in front of target. The sum of BST samples with dimension of 5x15 mm were fixed on holder and covered with polymer in the same time. An example of SEM and AFM images of PEI films on BST is presented in Figure 1 (the figure is in attached word file).

Point 3: Table 1 shows that the thickness of films deposited using 0.1 mbar decreases by increasing the Sr amount (x=0.5 to 0.75). However, the opposite behaviour occurs for the films deposited using 0.2 mbar. These results should be discussed.

Response 3: The thicknesses of samples were found to be between approximately 200 nm to 390 nm and varied with deposition parameters. The difference in thickness values can be explained by a couple of factors. First of all, the different laser absorption of targets because of different amount of dopant in BaTiO3 targets. The plasma plume resulted after interaction of laser with targets are different. On the other hand, a higher pressure of gas in the deposition chamber leads to confinement (confiner) of plasma plume. A raising in the gas pressure results in an increase of the fluorescence, sharpening of the plume boundary and the shock front, slowing of the plume relative to the propagation in vacuum, and a higher spatial confinement of the plume. In both cases, the thicknesses were lower at high pressure. A combination of these factors can lead to different values of thickness.  

Point 4: How many measurements were performed in the same device to analyse the repeatability of the response? Also, how many sensors were tested to examine the reproducibility of results?

Response 4: Thank you for your comment, the repeatability of the whole process, including here the functional properties of the thin films, is a very important factor to consider. In PLD deposition process, the biggest challenge is the variation of deposition parameters during the process. For example, a variation in oxygen flow or oxygen pressure during deposition will affect the properties of BST. Another problem is the variation of laser energy from pulse to pulse, and for good results this variation needs to be below 10%. A correct choice of deposition parameters and a good functionality of the PLD subsystems must be done in order to obtain correct properties of films.

In the section “2.1. Thin Films and IDT Deposition”, first paragraph, the following comment has been introduced:

  “For each composition of BST and oxygen pressure we deposited four probes in a row and we checked their morphological and structural properties before we proceeded with the deposition of gold electrodes, followed by gas response measurements.”

  In the section “3.4. SAW measurements for different gases”, second paragraph, the following comment has been introduced:

“For the sensors that yield a frequency response, we did around 10 measurements and we found the repeatability of the response had differences of below 3%, with a maximum difference in reproducibility of 5%.”

Even in the case of the samples with lower frequency response, because on each sample we deposited 3 or 4 gold electrodes, we tested some but without success.

In Figure 2 (figure is in doc file attached), cycles of gas response are presented. The x axis represents the time of response and is different for each of O2, CO2 and N2 gases.

Point 5: What is the proposed sensing mechanism that explains the higher response of BST50 0.2 mbar to CO2 and the selective detection of BST75 0.2 mbar considering the effect of the polymer layer?

Response 5: There is no response at CO2 or O2 for BST50 without polymer. For the N2 the frequency response is very noisy.  In the case of adding PEI on the surface of BST, the response became much clearer. PEI is a well-known polymer for gas detection and there are plenty of scientific paper with reports on PEI detection capability.  Bin Sun et al. explained the mechanism of PEI detection for CO2 and is based on Hard Soft Acid Base (HSAB) theory. The interaction of amino group of PEI with CO2 at room temperature forms carbonates, by a reversible reaction.

Point 6. There are some differences between the discussion and the Figures and Tables. For example, the authors mentioned noise measurements in Figure 7 (a), (c), and (e). However, this was observed in Figures 7(a), (b), and (c). Also, on page 12, the authors discussed the results of the BST75 sample, and the higher response of 0.183MHz mentioned is related to the BST50 sample, as observed in Table 1. The explanation of the results should be clear.”

Response 6: Thank to reviewer for carefully reading. We have corrected our mistakes in Figure 7.

Point 7: The highest response obtained was from the BST50 0.2 mbar sample (0.3 MHz frequency shift for CO2), which is different from the statement in the last sentence of the abstract. It should be corrected.

Response 7: We corrected the difference between abstract and text.

Point 8: Both Sr contents for the BTS50 samples were attributed to the sample prepared in 0.2mbar. It should be corrected (pag.7).

Response 8 Thank you for your observation. The text correction was done at page 7.                       

Point 9: References or a brief description of the sintering route used for target preparation should be included in section 2.1.

Response 9: We have introduced a reference for targets sintering route, reference number [39] in manuscript.

Point 10: The X-axis in Figures 6a-b should be labelled.

Response 10: We labelled the Figure 6 (a)-(b).

Point 11: The abbreviations QCM and FBAR on page 2 should be defined for a non-specialized audience.

Response 11: In page 2 we defined the abbreviation: film bulk acoustic resonators (FBAR) and quartz crystal microbalances (QCM)

Final remarks:

Indeed, the sensing measurements were conducted in pure gases and that was done because we did not know if our materials will respond or not to the presence of gases. Once the effect of the deposition parameters and the concentration of strontium in BST layer was evidenced, this opened a way for developing a suitable SAW sensor based on BaSrTiO3. The future work will be focused on detection of low concentration and specific gases in a mixture. All measurements were done at room humidity (~60%) and temperature (~250 C). The role of humidity is very important for final application (gas sensing in different medium) and this also represents a task for future work.

Reviewer 2 Report

This work presents surface acoustic wave devices for gas sensor applications by using BaSrTiO3 synthesized by means of pulsed laser deposition. They comprehensively investigated BaSrTiO3 thin-films through AFM, XRD, SEM, and UV-vis. Furthermore, they demonstrated surface acoustic wave devices to detect various gases including N2, CO2 and O2. This reviewer recommend acceptance for publication after considering following minor comments: 

(1) Please increase visibility of figures, for especially Figure 1, Figure 2 scale bar 1 um, Figure 5.

(2) Please provide eds chemical analysis to clarify the presence of BaSrTiO3. The authors have already measured the SEM, which can offer an eds mapping for each atomic dispersion.

(3) Please mention full name of PEI at its first appearance. 

(4) Regarding surface acoustic wave measurement, how can this approach distingush each gas? Would be possible to distingush target gas from mixed gases atmosphere? 

Author Response

Response to Reviewer 2

Comments and Suggestions for Authors

This work presents surface acoustic wave devices for gas sensor applications by using BaSrTiO3 synthesized by means of pulsed laser deposition. They comprehensively investigated BaSrTiO3 thin-films through AFM, XRD, SEM, and UV-vis. Furthermore, they demonstrated surface acoustic wave devices to detect various gases including N2, CO2 and O2. This reviewer recommend acceptance for publication after considering following minor comments:

Point 1: Please increase visibility of figures, for especially Figure 1, Figure 2 scale bar 1 um, Figure 5.”

Response 1: The visibility of figures was increased.

Point 2. Please provide eds chemical analysis to clarify the presence of BaSrTiO3. The authors have already measured the SEM, which can offer an eds mapping for each atomic dispersion.”

Respunse2: Thank you for your comment. Due to the high differences in the functional properties of the perovskite thin films induced by the stoichiometry deviations within the sample, we are now performing a study on the fine stoichiometry measurements using Secondary Ion Mass Spectroscopy (SNMS/SIMS) and XPS techniques.  We consider that the EDS technique does not provide the required accuracy both qualitatively and quantitatively terms, needed to fully reveal for such tremendous effects on the functionality of the perovskite thin films.  We have performed some EDS analysis, presented in attached figure. All elements from BaSrTiO3 and Al2O3 substrates are presented.

The element counts are listed below for sample of BST75 deposited at 0.2 mbar of oxygen:

Element

Weight %

Atomic %

Error %

C  K

45.91

56.95

8.02

N  K

22.35

23.78

10.27

O  K

10.65

9.92

10.22

Al K

14.65

8.09

3.48

Sr L

3.22

0.55

1.85

Ba L

1.45

0.16

8.31

Ti K

1.79

0.56

2.46

Table 1. The element list and percentages resulted from EDX analysis for BST75_0.2 mbar

Point 3: Please mention full name of PEI at its first appearance.

Response 3: Thank you for your comment, the full name of PEI has been mentioned.

Point 4: Regarding surface acoustic wave measurement, how can this approach distinguish each gas? Would be possible to distinguish target gas from mixed gases atmosphere?”

Response 4: In order to distinguish between gases in a mixed atmosphere, a matrix of sensors must be made, each being deposited with different sensitive materials. For example, for sensors deposited with PEI sensitive materials, the sensor had different responses to the gases tested (N2, CO2 and O2). In the next stage, we will develop a matrix of sensors, for detecting between several gases.

Reviewer 3 Report

In this manuscript, Nicoleta Enea et al report a new method to fabricate layered structures gas sensors by using laser deposition techniques. The laser method seems quite effective, and the performance of the resulting devices is well demonstrated. Some minor concerns should be addressed before publishing.

1.     In the introduction part, the differences and limitations of BAW and SAW are not discussed. Is laser fabrication also applicable to BAW?

2.     The mechanism of using BST for different gas sensing should be discussed.  What parameters are changing when BST is exposed to different gases?

3.     What is the advantage of the laser deposition technique for BST fabrication? Will laser fabrication reduce the preparation time? Cost? Improve the accuracy? or sensitivity? Is there any challenge to using laser technology in mass production?

4.     In the discussion part, there is no discussion of optimizing laser parameters for fabrication. Will the power influence the performance? Will the different laser generation methods affect the composition of BST? Will the electrode material affect the performance of devices?

5.     For BST fabrication, will the random heat generated by the laser affect the stability of fabrication?

6.     Does the sensor have the ability to identify the specific gas (eg CO2) in a mixture of gases?

Author Response

Response to Reviewer 3

Comments and Suggestions for Authors

In this manuscript, Nicoleta Enea et al report a new method to fabricate layered structures gas sensors by using laser deposition techniques. The laser method seems quite effective, and the performance of the resulting devices is well demonstrated. Some minor concerns should be addressed before publishing.

Point 1: In the introduction part, the differences and limitations of BAW and SAW are not discussed. Is laser fabrication also applicable to BAW?

Response 1: A discussion regarding BAW and SAW was added in Introduction.

There are numerous papers in the literature, including review papers, that compare these two types of sensors (BAW and SAW) and in which their differences and limitations are discussed. Basically, SAW sensors use surface acoustic waves, and BAW use bulk or volume acoustic waves. In BAW devices, acoustic waves propagate through the bulk structure of the device active layer. The BAW structure consists of the piezoelectric thin film sandwiched between two metal electrodes and the acoustic waves propagate through the bulk structure of the device active layer. The most commonly used bulk acoustic wave (BAW) devices are the thickness shear mode (TSM) resonator. The TSM features simplicity of manufacture, ability to withstand harsh environments, temperature stability, and good sensitivity to additional mass deposited on the crystal surface. As a result of its shear wave propagation component, the TSM resonator is also capable of detecting and measuring liquids, making it a good candidate for a biosensor. Unfortunately, these devices have a low mass sensitivity. Typical TSM resonators operate between 5–30 MHz. Making very thin devices that operate at higher frequencies can increase the mass sensitivity, but thinning the sensors beyond the normal range results in fragile devices that are difficult to manufacture and handle. In general, the sensitivity of the sensor is proportional to the amount of energy that is in the propagation path that is being perturbed. BAW sensors typically disperse the energy from the surface through the bulk material to the other surface. This distribution of energy minimizes the energy density on the surface, which is where the sensing is done. Surface wave acoustic sensors, conversely, focus their energy on the surface, making them typically more sensitive sensors. The SAW sensor, typically consist of a piezoelectric thin film with two metal electrodes on the same side of the piezoelectric substrate. The acoustic wave propagates on the surface of the piezoelectric surface between these two electrodes (input and output).

Laser methods are applicable for BAW sensors because we can make the deposition of the piezoelectric substrate through these methods and we can ensure and control the necessary properties of the substrate for the operation of the BAW sensors.

Point 2: The mechanism of using BST for different gas sensing should be discussed.  What parameters are changing when BST is exposed to different gases?

Response2: In the section “3.4. SAW measurements for different gases”, second paragraph, the following comment has been added:

“The mechanism of gas sensing was explained by Wang et al. for the oxide materials.

When gas molecules are interacting with surface of BaSrTiO3 the charge state is altered and the conductivity of sample is changed. The changing in conductivity of layer leads to change in SAW sensor response. The selectivity of SAW sensors is solely dependent upon the material properties of the sensing layer. When a SAW sensor is exposed to a mixture of various gases, it absorbs or reacts to them differently, so the strength of the sensor output is different for each component. The rate of layer-analyte interaction and hence the response time of a SAW sensor is affected by several factors. In the case of mass-based sensing layers, the response and recovery times of a SAW sensor mainly depend upon the rate of diffusion of the absorbed mass into the film and to the piezoelectric substrate and back to the film surface. The sensors with thinner sensing layers usually have faster response kinetics. With decreasing the thickness, the gas diffuses in and out of the film rapidly, so that the time required to reach to the equilibrium decreases. Thin sensing layers thus result in rapid responses to gases.

Point 3: What is the advantage of the laser deposition technique for BST fabrication? Will laser fabrication reduce the preparation time? Cost? Improve the accuracy? or sensitivity? Is there any challenge to using laser technology in mass production?

Response 3:

What is the advantage of the laser deposition technique for BST fabrication?

Advanced film fabrication techniques such as chemical vapor deposition, sputtering, vacuum evaporation, selective laser sintering have been utilized to fabricate high quality BaTiO3 thin films. High surface roughness (which increases optical loss) and stoichiometry deficiency are associated with those methods. Pulsed-laser deposition technique offers one of the best methods for the growth of complex oxide films and especially perovskites. The advantage comes from the possibility to careful control the deposition parameters and to obtain crystalline thin films without any post deposition treatments. By controlling multiple deposition parameters in the PLD process, for example gas pressure, repetition rate, temperature, target-substrate distance leads to obtaining perovskite thin films with different properties.

Will laser fabrication reduce the preparation time? Cost?

Yes, for example if we compare it with other techniques, PLD is less time consuming and costs less.

Improve the accuracy? or sensitivity?

Yes, slightly changing the thin film material properties (varying the deposition parameters) can lead improve accuracy and sensitivity of the final device.

The sensitivity of a sensor can be significantly changed by using materials with different grains sizes.  Ansari et al.  evidenced this behaviour for nanocrystalline SnO2. Lu et al.  reported a SnO2 -based sensor response to 500 ppm CO increases drastically if the particle diameter becomes smaller than 10 nm. These studies underline the importance of thin films crystalline state on gas response of oxide materials.

In our case for BST because we obtained nanocrystalline thin films with dimension of nanocrystals around 20-25 nm.  More studies regarding the influence of crystalline state of BST on accuracy or sensitivity in gas detection will be done in the future.

PLD systems are already available on the market for mass production. The large area PLD opens the possibility to obtain high quality coatings at lower costs. The wafer covered with thin films using PLD can have a diameter of 300 mm and, combined with techniques for metallization of photolithography on large area is possible to obtain tens of thousands of gas sensors in a single technological chain.

Point 4: In the discussion part, there is no discussion of optimizing laser parameters for fabrication. Will the power influence the performance? Will the different laser generation methods affect the composition of BST? Will the electrode material affect the performance of devices?

Response 4: Firstly, we did an optimization of the laser parameter (number of laser pulses and the laser fluency) to obtain the BST samples with thickness of hundreds of nanometres but we did not present this work because, for gases response, we chose the thicker samples and the others unmeasured samples are not relevant for this study. The thicker samples have the same properties like bulk materials. Of course, the thinner samples can be very interesting for studying in these applications because there are some effects which can be exploited, such as high strained thin layer (up to 10-20 nm), with enhanced properties (ex: piezoelectric effect). The crystalline structure in a very thin layer can be easily deformed and the acoustic wave speed in a strained layer can be different and subsequently, the gas response can change. This study is of interest and could represent a future approach.

The materials used as electrodes must be carefully chosen. For example, we can use other metals (aluminium for example is a much cheaper material), but Al oxidizes (especially in rich oxygen environment) and the aluminium oxide can influence the electrical response. We used gold deposited by thermal evaporation for SAW electrodes because we wanted to minimize possible risks caused by the quality of electrodes.

Point 5: For BST fabrication, will the random heat generated by the laser affect the stability of fabrication?

Response 5: We did not fully understand the reviewer question, nevertheless, there is no random heat generated by laser.

In PLD process just the target is ablated (nanosecond regime ablation) by laser and the heat can appear on surface of target. To avoid the random heating phenomenon and ablating for the same area that will result in target damage, uneven ablation and unevenness in the thin film distribution on the surface of the substrate, the ceramic target is rotated and the laser beam is translated on surface, during the deposition process. The laser beam does not interact with the substrate or other mechanical parts in deposition chamber. The fabrication stability can be affected by the variation of the deposition parameters (fluctuation of oxygen, laser energy, etc) but those parameters can be fully controlled during the process.

Point 6: Does the sensor have the ability to identify the specific gas (e.g., CO2) in a mixture of gases?

Response 6: Multiple gas detection and identification is possible if several SAW sensors, each with a unique chemically specific coating, are placed in an array. Each SAW sensor will have a different output given the same vapor exposure [4] In order to do so, a matrix of sensors must be made, each being deposited with different sensitive materials. For example, for sensors deposited with PEI sensitive materials, the sensor had different responses to the gases tested (N2, CO2 and O2). In the next stage, we will develop a matrix of sensors, to be able to detect between several gases.

Round 2

Reviewer 1 Report

I am not completely satisfied with the authors' reply, but I recommend the publication of the manuscript.